# Online computer or therapist-guided cognitive behavioral therapy in university students with anxiety and/or depression: study protocol of a randomised controlled trial

Anke Klein [1,2] N E Wolters,[1,2] E J M Bol,[1] J Koelen,[1] L de Koning,[1] S S M Roetink,[1] J J van Blom,[1] T Pronk,[1,3] Claudia van der Heijde,[2] Elske Salemink,[1,4] Felix Bolinski,[5] Heleen Riper,[5] Eirini Karyotaki [5] Pim Cuijpers [5] S Schneider,[6] Ronald M Rapee,[7] Peter Vonk,[2] Reinout W Wiers[1]

For numbered affiliations see end of article.

**Correspondence to**
Dr Anke Klein; A.M.klein@uva.nl

## ABSTRACT

**Introduction** Emerging adulthood is a phase in life that is associated with an increased risk to develop a variety of mental health disorders including anxiety and depression. However, less than 25% of university students receive professional help for their mental health reports. Internet-based cognitive behavioural therapy (iCBT) may entail useful interventions in a format that is attractive for university students. The aim of this study protocol is to test the effectiveness of a therapist-guided versus a computer-guided transdiagnostic iCBT programme with a main focus on anxiety and depression.

**Methods and analysis** University students with anxiety and/or depressive symptoms will be randomised to a (1) 7-week iCBT programme (excluding booster session) with therapist feedback, (2) the identical iCBT programme with computer feedback only or (3) care as usual. Participants in the care as usual condition are informed and referred to conventional care services and encouraged to seek the help they need. Primary outcome variables are self-reported levels of anxiety as measured with the General Anxiety Disorder-7 and self-reported levels of depression as measured with the Patient Health Questionnaire-9. Secondary outcomes include treatment adherence, client satisfaction, medical service use, substance use, quality of life and academic achievement. Assessments will take place at baseline (t1), midtreatment (t2), post-treatment (t3), at 6 months (t4) and 12 months (t5) postbaseline. Social anxiety and perfectionism are included as potentially important predictors of treatment outcome. Power calculations are based on a 3 (group) × 3 (measurement: pretreatment, midtreatment and post-treatment) interaction, resulting in an aimed sample of 276 participants. Data will be analysed based on intention-to-treat and per protocol samples using mixed linear models.

**Ethics and dissemination** The current study was approved by the Medical Ethics Review Committee (METC) of the Academic Medical Centre, Amsterdam, The Netherlands (number: NL64929.018.18). Results of this trial will be published in peer-reviewed journals.

**Trial registration number** NL7328.

### Strength and limitations of this study

► The current study protocol describes a randomised controlled trial (RCT) focused on university students with anxiety and/or depressive symptoms.
► A transdiagnostic internet-based cognitive behavioural therapy programme, with either therapist guidance or computerised guidance, is compared with care as usual (three-arm RCT).
► The procedure of the trial is developed to make it easily accessible and anonymous, and the full procedure is online.
► The therapists are not fully blinded to the treatment condition.
► The post-treatment outcomes rely on self-reports of the participants.

## BACKGROUND

Emerging adulthood is a crucial stage in life that includes the transition from the teenage years to full-fledged adulthood (18–25 years). It is a phase during which many young adults attend university to receive education and training to obtain a degree that will be the basis for their adult working life.[1] It is also the phase that is associated with an increased risk of developing a variety of mental health disorders such as anxiety and depression,[2–4] whereby comorbidity is the rule rather than the exception.[5] Recent studies found that between 16% and 46% of all students suffer from mental health disorders.[2 6] Moreover, these problems have been associated with poorer academic performance, academic dropout, decreased labour market functioning and poorer social and health outcomes.[6–10] Early detection and effective intervention are, thus, crucial with regard to the long-term prognosis of mental

disorders, especially as untreated mental disorders have a mean duration between 4 and 23 years.[11]

Online treatment programmes may be useful as an intervention in a format that is particularly attractive for university students, given the fact that less than 25% of students with mental health problems use available face-to-face services.[12] University students might avoid help due to several reasons, including fear of stigmatisation, lack of perceived urgency, preference to deal with it themselves, threat to identity formation and goal setting and/or scepticism regarding treatment effectiveness.[13 14] This group may be more likely to seek help if an effective, more flexible, easily accessible and anonymous intervention strategy is provided. A scoping review by Boydell *et al*[15] concluded that internet therapies have (untapped) potential to provide better information about mental health and may improve the cost efficacy of mental health services. Moreover, their study indicated that young people, in general, prefer eHealth treatments relative to face-to-face services.

Several studies demonstrated that the effectiveness of guided internet-based cognitive behavioural therapy (iCBT) programmes for anxiety and depression is comparable to conventional (face-to-face) CBT (for meta-analyses, see Carlbring *et al* and Cuijpers *et al*[16 17]). There is also some evidence for the effectiveness of internet interventions for young adults and university student populations with various mental health problems, including depression, anxiety, eating disorders, stress, alcohol and sleep problems[18–22]. For example, a recent meta-analysis[20] focused on university students with mental problems, who self-selected a psychological online intervention (guided/unguided). They found small intervention effects for depression ($g$=0.18, 95% CI (0.08 to 0.27) and anxiety ($g$=0.27, 95% CI (0.13 to 0.40)) in university students. It should be noted, however, that the effects on anxiety were not significant anymore after adjusting for publication bias. Harrer and colleagues[20] concluded that clearly more research is needed to explore ways to increase treatment effectiveness for university students and to study predictors of treatment outcome.

The current study includes a group of university students and focuses on two of the most common mental disorders in this group, namely, anxiety and depression.[2] As comorbidity is high, we will use a transdiagnostic iCBT intervention, which has the advantage to target both disorders at the same time and thereby might increase treatment effectiveness (for a systemic review and meta-analysis, see Newby *et al*[23]). Moreover, iCBT programmes can be provided with or without additional personal guidance. Guidance usually implies written or verbal feedback by the therapist to the patient's homework assignments and other forms of (safe) communication (eg, chat, telephone) within an online environment with varying degrees of interactive features.[24] Both approaches (with and without personal guidance) have shown to be efficacious, although programmes with (therapist) guidance generally report a higher effect size (for meta-analyses,

see Cuijpers *et al* and Richards and Richardson [17 25]), even though it should be noted that a recent meta-analysis[20] found no difference between personal or no personal guidance. To date, little is known about the effectiveness for therapist-guided versus a computer-guided version of the same intervention tested *within* the same study when applied to a university student sample with mental health concerns[20 26 27]. This could also have important health economic implications considering the additional cost of human support. Therefore, the current study tests this transdiagnostic iCBT for anxiety and depression in three groups; a therapist-guided online self-help version, a computer-guided version of the same programme and care as usual (CAU). Primary outcomes were anxiety and depression; secondary measures were included to evaluate treatment adherence and satisfaction, academic achievement and medical service use.

Internet-based programmes, either therapist-guided with personal feedback or computer-guided with automated feedback, might suit some individuals better than others. It is important to focus on possible treatment predictors (eg, including baseline symptoms severity, gender, personality factors and comorbidity), as this may help us understand more about the heterogeneity of treatment response.[20 28] Specially, a recent meta-analysis[20] found limited effectiveness on anxiety in university students in their meta-analysis. The current study will focus on the possible role of social anxiety and perfectionism, as there is some evidence that adults and children with social anxiety might benefit less from general CBT programmes (for meta-analyses, see Hudson *et al* and Norton and Price[29 30]). Likewise, self-critical perfectionism is a transdiagnostic personality factor that is generally associated with worse treatment outcome, including CBT.[31] Importantly, individuals with self-critical perfectionism have poorer social relationships, which in turn negatively impacts therapeutic progress.[32]

The aim of this study is to test the effectiveness of a therapist-guided versus a computer-guided transdiagnostic iCBT programme with a focus on anxiety and depression. Based on previous studies in the general population,[17] we hypothesise that the therapist-guided iCBT and computer-guided iCBT conditions are both significantly more effective than CAU. In addition, we expect that the therapist-guided iCBT condition is significantly more effective than the computer-guided iCBT condition; an effect we expect to be mediated by higher treatment adherence.

## METHODS
### Study design
The current study includes a randomised controlled trial (RCT) design with three arms, consisting of (1) a therapist-guided transdiagnostic iCBT programme with personalised feedback, (2) a computer-guided version of the same CBT programme with automated feedback and (3) CAU. We used the Standard Protocol Items:

Recommendations for Interventional Trials reporting guidelines for this study[33]; see online supplemental appendix A.

## Eligibility criteria

All students scoring 16 or higher on the Center for Epidemiological Studies Depression Scale (CES-D)[34] and/or a score of 5 or higher on the General Anxiety Disorder-7 (GAD-7)[35] were included in the study. There is no higher end cut-off. Participants are excluded from the study if they (1) are younger than 16 years of age, (2) currently receive psychological treatment (as described by the American Psychological Association (APA) dictionary) for anxiety and/or depression, (3) do not have access to a stable internet connection for the next 2 months, (4) if they meet the diagnostic criteria for (recent or current) psychotic disorder according to the Mini International Neuropsychiatric Interview (MINI),[36] (5) if they are at active high risk for suicide based on the MINI and (6) if they do not give informed consent. Participants meeting exclusion criteria 4 and/or 5 are referred to local healthcare facilities.

## Recruitment

All participants were recruited at the University of Amsterdam (UvA), The Netherlands. The UvA is a public research university located in Amsterdam. It is one of the largest research universities in Europe with 31 186 students enrolled in 2018 and includes seven faculties: Humanities, Social and Behavioural Sciences, Economics and Business, Science, Law, Medicine and Dentistry. Recruitment for the study is done in two phases: a first broad screening phase for all university students studying at the UvA and a second more specific screening of eligible students who showed interest to participate. In the first broad screening phase, all students enrolled at the UvA receive an invitation via email that are sent centrally from the study platform. In addition, study advisors and study counsellors of the UvA are informed about the study and are asked to refer students who are interested in participating in the study to the research team, so that they can be invited by email.

This first broad-screening invitation email contains general information about the study and a unique link to fill out the screening questionnaire. This screening questionnaire takes approximately 20 min to complete and also includes other questionnaires related to the mental health of university students. This screening questionnaire was developed in collaboration with other universities to collect multicentre data. Participants who click on the invitation link are referred to the project platform. Here, they find an information letter and are asked to give online informed consent to the research team. This information letter explicitly states that participation is voluntary and that participants can withdraw at any time without consequences. Next, participants are asked to create an account after which they are directed to the screening questionnaire. Following the original invitation, two reminder emails are sent 1 and 2 weeks after the first invitation. As the study runs for three academic years (2019–2021), participants are asked at the beginning of the screening if they consent to being invited again later. Moreover, participants can also indicate that they do not want to receive any emails again.

In the second more specific screening phase, university students who score above the cut-off on either anxiety (GAD-7) and/or depressive symptoms (CES-D) receive an email with an online information brochure and informed consent document[34 35]; see online supplemental appendix B. After informed consent is given by the participant, they are called by a trained clinical research assistant to check all of the inclusion and exclusion criteria. A telephonic diagnostic interview (MINI)[36] is administered to establish possible diagnoses with respect to mood and anxiety disorders, bipolar disorder, psychosis and suicidal ideation (not fulfilling the criteria for an anxiety and/or mood disorder is not an exclusion criterion).

## Randomisation, blinding and treatment allocation

Participants are randomly assigned to either the therapist-guided online intervention, the computer-guided online intervention or CAU (control) condition (1:1:1 allocation ratio) directly following the baseline measurement. Randomisation is stratified by gender and anxiety and/or depressive symptoms (based on the cut-off scores on the screening questionnaire). Sequence generation is based on computer-generated random numbers and are allocated by an automatic system.

Due to the nature of the intervention, participants and researchers cannot be completely blinded to the assigned treatment condition, though note that the measurements in the intervention are blinded. Researchers are aware of the participants' condition because they provide participants with feedback in the therapist-guided condition. In the computer-guided condition, feedback is automatically generated by the system, but therapists do send reminders to this group as part of the intervention.

## Intervention

The online transdiagnostic intervention that is used in this study, 'ICare Prevent', was originally developed by Weisel et al[37] for the general German-speaking population and translated and adapted by Bolinski et al[38] and Karyotaki et al[39] for a Dutch undergraduate student population into Dutch and English. The intervention was based on other effective protocols[40–43] and can be seen as a variant of an iCBT protocol for which several studies have proven its effectiveness. The intervention is provided to the participants through the eHealth platform Minddistrict. This platform enables researchers and healthcare providers to provide digital therapy to their participants and patients. On this platform, e-coaches create a personal account for each participant directly following the baseline assessment and randomisation. Once the personal account has been created, the participant receives an email with information how to activate the account.

Once the account is activated, the participant can start with the intervention. The platform monitors the progress of the participants, e-coaches provide support for the participants in the therapist-guided treatment condition and respond to questions from the participants with the messaging function. Data processing and storage are in accordance with the ISO 27000 and NEN 7510 norms. A data processing agreement that complies with the European General Data Protection Regulation (GDPR) has been signed between eHealth provider Minddistrict and the UvA.

The intervention is based on basic cognitive behavioural therapy principles for anxiety and depression with components of psychoeducation, including online exercises and home assignments. The intervention consists of seven regular sessions (45–60 min per session) and one booster session (4 weeks after the completion of the last session). From the second session onwards, participants are able to follow eight additional optional modules based on their personal needs, including sleep, perfectionism, alcohol use, reduce rumination, self-worth, acceptance, appreciations and rest and relaxation. They can decide to choose one additional module per session, and they are free to repeat this module or choose other modules in later sessions. In sessions 5 and 6, users decide to follow either content directed at changing negative cognitions or exposure to feared situations. Participants are free to decide when they would like to start a treatment session and if they want to do an additional module, but it is advised to do at least one and no more than two treatment sessions per week. For a full description of the intervention, see Karyotaki *et al.*[39]

### Therapist-guided intervention versus computer-guided intervention

During the intervention, participants in both conditions receive online support in the form of messages or notifications in the messaging function (e-mails). In both conditions, participants receive up to 3 weekly reminders via the messaging function when they are inactive. Moreover, participants in both conditions can use the messaging function to ask questions. All participants can ask technical or usability related questions. There is one key difference between the two conditions: Participants in the computer-guided iCBT condition receive automatically generated feedback messages (see online supplemental appendix C for an example). In contrast, in the therapist-guided condition, the e-coaches (trained psychologists on minimal Bachelor (BA) level) provide detailed feedback based on participants' output of the sessions, and this feedback is displayed at the bottom of the session they completed (see Appendix C for an example). The coach spends approximately 30 min on providing feedback per session and intends to give feedback within five working days after the session is completed by the participant. During the trial, training and weekly supervision for the e-coaches who are guiding the participants are provided by a licensed mental healthcare psychologist.

### Care as usual

Participants in the CAU condition are informed and referred to conventional care services and encouraged to seek the help they need. Medical services used by the participants during the RCT are monitored through self-report questions at t3, t4 and t5. Participants in the CAU condition are assessed at the same time points as the two intervention conditions including the weekly questionnaires during the first 7 weeks.

### Suicide risk monitoring

All participants receive an information brochure with all relevant contact information as part of the information package they receive prior to the study. In addition, the brief version of the PHQ-9, the PHQ-4[44] is administered before each treatment session or weekly via email (CAU). Suicide risk is monitored with the help of a suicide risk protocol that was created specifically for this study (Klein *et al Suicide protocol).* Suicide risk is monitored using item 3 ('feeling down, depressed or hopeless') of the PHQ-4 and item 9 ('thought that you would be better off dead, or of hurting yourself') of the Beck Depression Inventory scale.[45] If deemed necessary, participants are called and referred to appropriate services, including their General Practitioner (GP) or 113zelfmoordpreventie (online platform for suicide prevention). In addition, the participant also receives a pop-up message on the questionnaire page with relevant contact information right after filling in the questionnaire (ie, where they can find help if needed with relevant contact information).

### Assessments

The RCT includes five assessment points: (1) a baseline assessment including a diagnostic interview and questionnaires, (2) a midtreatment questionnaire, 5 weeks after baseline, (3) a post-treatment questionnaire, 8 weeks after baseline and (4) follow-up measurements, 6 months and (5) 12 months after baseline (figure 1). In addition, prior to each treatment session, or weekly in the CAU condition, participants fill out a short online questionnaire. This questionnaire briefly assesses symptom severity and monitors suicidal ideation.

### Primary outcomes

The Generalised Anxiety disorder-7 (GAD-7)[35] and the Patient Health Questionnaire-9 (PHQ-9)[46] are the primary outcomes in the study and assess respectively *anxiety and depressive symptoms* on a continuous scale. The 7-item GAD is a self-report questionnaire measuring anxiety symptoms. All items are scored on a scale from 0 ('not at all') to 3 ('nearly every day'), with total scores ranging from 0 to 21. A higher score on the questionnaire is associated with a more severe experience of anxiety symptoms. The GAD-7 questionnaire has good psychometric properties including a good test retest reliability (ICC=0.83)[35] and a good internal consistency ($0.79 < \alpha < 0.91$).[47] The PHQ-9 questionnaire is used to screen for self-reported symptoms of depression. The nine items are scored on a 0–3 scale,

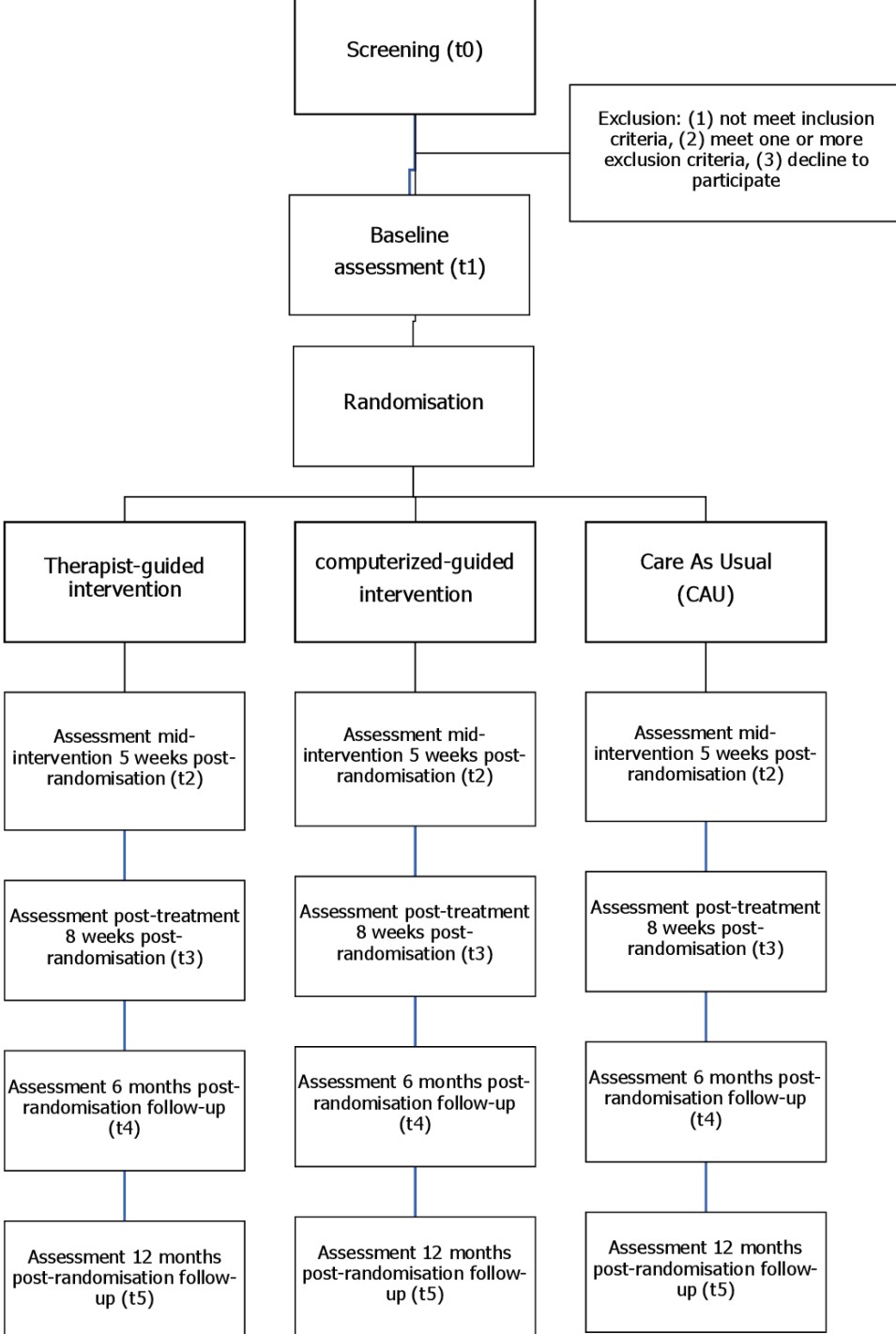

**Figure 1** Flow of the study.

resulting in a total score between 0 and 27. A higher score on the scale indicates a more severe experience of depressive symptoms. The PHQ-9 is a valid and reliable tool with good sensitivity (0.88) and high specificity (0.94)[48 49] (see table 1 for an overview of all questionnaires on all time points).

### Secondary outcomes

*Alcohol* and *drug use* are measured with respectively the Alcohol Use Disorders Identification Test-Concise

(AUDIT-C)[50 51] and the Drug Abuse Screening Test – 10 items (DAST-10).[52] The AUDIT-C is a brief screening tool for identifying early and/or heavy alcohol use. The scale consists of 3 items, all scored on a 0–4 scale, resulting in a total score ranging from 0 to 12. The DAST-10 consists of 10 items. Item responses are 'yes' or 'no'. Higher scores indicate a more severe level of drug abuse problems. The DAST-10 reports high validity ($\alpha=0.94$) and reliability in a broad range of settings and populations.[53]

**Table 1** Overview of the measures separately for each assessment point

| | Questionnaire | Assessment points | | | | | |
| --- | --- | --- | --- | --- | --- | --- | --- |
| | | Baseline (t1) | Presessions | Midtreatment (t2) | Post-treatment (t3) | 6 months follow-up (t4) | 12 months follow-up (t5) |
| Anxiety | GAD-7 | X | X* | X | X | X | X |
| Depression | PHQ-9 | X | X* | X | X | X | X |
| Social anxiety | SIAS-6 | X | | X | X | X | X |
| Social anxiety | Mini-SPIN | X | X | X | X | X | X |
| Alcohol use | AUDIT-C | X | | X | X | X | X |
| Drug use | DAST-10 | X | | X | X | X | X |
| Quality of life | EQ-5D | X | | | X | X | X |
| Objective academic achievement | † | X | | | X | X | X |
| Subjective academic achievement | PSS-WIS | X | | | X | X | X |
| Perfectionism | DEQ-SCP (short) | X | | | | | |
| Suicide risk | BDI-II (item 9) PHQ-4 (item 3) | | X | | | | |
| Medical service use | TiC-P | | | | X | X | X |
| Satisfaction with intervention | CSQ-8 | | | | X | | |

*PHQ-4, brief anxiety and depression measure to monitor symptoms.
†Self-developed questionnaire.
AUDIT-C, Alcohol Use Disorders Identification Test-Concise; BDI-II, Beck Depression Inventory scale –II; CSQ-8, Client Satisfaction Questionnaire—eight items; DAST-10, Drug Abuse Screening Test—ten items; DEQ-SCP, Depressive Experiences Questionnaire - Self-Critical Perfectionism; ECS-R-SF, Revised Experiences in Close Relationships—Short Form; EQ-5D-5L, EuroQol-5D-5L; GAD-7, General Anxiety Disorder-seven items; Mini-SPIN, Mini-Social Phobia Inventory; PHQ-4, Patient Health Questionnaire—four items; PHQ-9, Patient Health Questionnaire-nine items; PSS-WIS, Presenteeism Scale for Students—Work Impairment Scale; SIAS-6, Social Interaction Anxiety Scale—six items; TiC-P, Treatment Inventory of Costs in Patients.

*Quality of life* is measured with the EuroQol-5D-5L scale (EQ-5D-5L).[54] The EQ-5D-5L assesses health-related well-being on five dimensions: mobility, self-care, ordinary activities, discomfort and mood state. The sixth item of the EQ-5D-5L is a VAS scale, ranging from 0 to 100, asking the participant to rate their health perception at that moment. The EQ-5D-5L has high levels of acceptability and sensitivity.[55 56]

*Academic achievement* is measured with objective and subjective measures. Students are asked to report their number of European Credit Transfer System as an objective measure of their academic performance. Additionally, the work impairment subscale of the Presenteeism Scale for Students (PSS-WIS)[57] is administered to assess subjective academic achievement on a 5-point Likert scale ranging from 'always' (0) to 'never' (5). The PSS-WIS is a valid (ICC=0.88 (95% CI p<0.001)) and reliable (test–retest reliability r=0.80, p<0.001) self-report measure to screen for impaired work performance due to (mental) health problems.[57]

*Medical service use* is measured with two items from the Treatment Inventory of Costs in Patients with mental disorders.[58] The first item includes the frequency of contact with conventional care services (eg, general practitioner, study advisor, psychologist, medical specialist). The second item includes the use of medication during the length of the treatment period.

*Client satisfaction with treatment* is measured with the Client Satisfaction Questionnaire—eight items (CSQ-8).[59 60] The CSQ-8 consists of 8 items scored on a 1 ('quite dissatisfied') to 4 ('very satisfied') scale, with total scores ranging between 8 and 32. The CSQ-8 is a standardised satisfaction measure reporting very good internal consistency (α=0.83–0.93) and high validity.[61]

Finally, following previous studies (for a systematic review),[62] *treatment adherence* is measured by tracking the activities in Minddistrict. We collect the total number of modules completed, time spent per module and the number of times the participants log into the Minddistrict platform.

### Predictors of treatment outcome

*Social anxiety* is measured with the Social Interaction Anxiety Scale-6 item (SIAS-6)[63] and the brief Mini-Social Phobia Inventory (Mini-SPIN).[64] The brief Mini-SPIN[64] is included to screen for generalised social anxiety disorder. The scale shows strong sensitivity (93.8%) and good internal consistency (α=0.85).[65 66] Besides the administration on t1–t5, the Mini-SPIN is administered before each treatment session, or weekly via email (CAU), to monitor the social anxiety status of the participant.

*Perfectionism* is measured with the short version of the Self-Critical Perfectionism subscale of the Depressive Experiences Questionnaire (DEQ-SCP).[67] The subscale

consists of 12 items. The DEQ-SCP scale is a simple, valid scale demonstrating good psychometric properties, in both clinical and student samples.[68]

Besides the inclusion of social anxiety and perfectionism, we include several variables to study possible predictors of treatment outcome including demographic variables (ie, age, gender, study direction, faculty, national/international student), baseline severity, personality factors, number of sessions completed/adherence rate and comorbidity for explorative analyses.

## Statistical analyses
### Sample size calculation
The power analysis is based on the mixed factors 3×3 interaction difference between the three treatment arms and three time points (premeasurement, midmeasurement and postmeasurement). We anticipate a small interaction effect (Cohen's f=0.1) at post-treatment based on a recent meta-analysis that focused on the effectiveness of therapist-guided versus computer-guided psychotherapy in treating depression and anxiety.[17] We determined a statistical power (1-ß) of .8 and corrected alpha of α=0.05. With a two-tailed hypothesis, we need 204 participants per intervention group to be able to detect a statistically significant result. Previous literature has shown that web-based interventions have a relatively high reported range of drop-out in university students, varying between approximately 30% and 40% posttreatment.[20] Calculating with a dropout rate of 35%, the minimum required total sample for the RCT is $N_{total}$=276 participants (with a ratio of 1:1:1), with 92 participants in each group.

### Clinical analyses
Mixed Linear Model analyses will be conducted with SPSS and R.

To get a more realistic picture of the true efficacy of treatment, completer analyses will also be performed for those participants who adhere to the protocol.

### Predictor analyses
Predictors of treatment outcome will be investigated on an exploratory basis with predefined predictors: social anxiety and perfectionism, baseline symptoms severity, gender, personality factors and comorbidity. Analyses will be conducted using regression analyses, with interaction effects and structural equation modelling.

## Patient and public involvement
A focus group of university students helped with the design of the study. This group also helped with writing the consent letters, the website texts and checked the clearness of all instructions on the online platform. All university students of the UvA will receive an email with the (anonymised) results at the group level when the project is finished.

## DISCUSSION
Even though many university students face mental problems, only relatively few students use available services.[12] iCBT programmes are easily accessible, flexible and relatively anonymous.[24] Hence, they might be a good alternative for reaching out to this group[14]. The current study aims to compare the effectiveness of online interventions for elevated levels of anxiety and/or depression in university students enrolled at the UvA: a therapist-guided transdiagnostic iCBT programme, a computer-guided version of the same programme and CAU. Several moderators are included that may potentially amplify or attenuate treatment response and so may provide new knowledge on effectiveness for different subgroups.

Strengths of this study are the use of a transdiagnostic iCBT programme, its large sample size, and the generalisability of the findings since exclusion criteria are minimal (eg, there are almost no exclusion criteria concerning comorbid disorders; we created a Dutch and English version, so that both national and international student can participate). Moreover, the current iCBT programme is flexible, because participants can complete sessions at their own pace and it includes separate modules focusing on related problems (eg, stress, sleep, alcohol use). This flexibility has both strengths and limitations. On the one hand, participants will not feel pressured, they can work on a session whenever they have time do to so, and they have freedom in choosing additional modules that fit their situation. On the other hand, this might result in a different therapeutic process for each participant, which makes it more difficult to compare the different groups. Also, this flexibility makes it more difficult to investigate the effective components of the programme. One way to investigate this is to follow the course of the symptoms by analysing the presession questionnaire. Some participants might find it difficult to stay motivated or decide between the different treatment module options. Therefore, one of the limitations of this study is a potentially high degree of drop out, especially in the computer-guided programme and in the CAU condition.[25] To minimise drop out, participants receive up to 3 weekly reminders via the messaging function when they are inactive. In addition, we provide all participants with technical support and suicide risk monitoring. Also, there may be a potential confound of 'feedback length', as the feedback in the therapist-guided condition is textually longer than the automated feedback in the computer-guided condition. Due to the large sample size but limited human resources, and our priority to maintain the privacy of participants, all measurements are self-reported and may, therefore, be more subjective than clinician-rated reports. Moreover, participants may not report certain information, for example, whether they receive face-to-face treatment or counselling for their complaints during the current study. Finally, our automated study design allows for blinded assessments throughout the intervention, which on the downside may lead to some assessments not being aligned with the

intervention phase (eg, 'midtreatment' questionnaires being completed before session 4 has been reached.)

In sum, the current trial is unique as it compares two versions of the same transdiagnostic iCBT intervention; a therapist-guided versus a computer-guided version in a large sample that includes a 1-year follow-up. This study will, therefore, improve our understanding of the need of individualised feedback for computer-based interventions. The findings will significantly add to the knowledge on the treatment of depression and anxiety in university students and will give insight into treatment predictors/moderators that may determine why some students benefit and others do not.

## ETHICS AND DISSEMINATION

The current study was approved by the Medical Ethics Review Committee (METC) of the Academic Medical Centre, Amsterdam, The Netherlands (number: NL64929.018.18) and follows all Dutch ethical legislations. Any unintended effects will be reported to the METC committee. All procedures are in accordance with the latest version of the Helsinki declaration and comply to the GDPR rules (2018). All protocol changes will be reported to the ethical committees, and trial participants per email. Active informed consent from participants is asked for the entire procedure. Results of this trial will be published in peer-reviewed journals.

**Author affiliations**
[1]Addiction Development and Psychopathology (ADAPT)-lab, Developmental Psychology, Department of Psychology, University of Amsterdam, Amsterdam, The Netherlands
[2]Department of Research, Development and Prevention, Student Health Service, University of Amsterdam, Amsterdam, The Netherlands
[3]Behavioural Science Lab, Faculty of Social and Behavioural Sciences, Universiteit van Amsterdam, Amsterdam, Noord-Holland, The Netherlands
[4]Social and Behavioral Sciences, Utrecht University, Utrecht, The Netherlands
[5]Department of Clinical Neuro- and Developmental Psychology, Amsterdam Public Health Research Institute, Vrije Universiteit Amsterdam, Amsterdam, Noord-Holland, The Netherlands
[6]Clinical Child and Adolescent Psychology, Ruhr University Bochum, Bochum, Germany
[7]Centre for Emotional Health, Macquarie University, Sydney, New South Wales, Australia

**Acknowledgements** We would like to thank the student psychologists, the student deans, colleagues of academic affairs, the colleagues from the Behavioural Science Lab (BSL) the student advisory boards who were involved in the focus group from the University of Amsterdam and M. van der Hoff and R. Atteveld for their help with the design of the project.

**Contributors** RWW (PI), PV (PI), AK (Co-PI) and CvdH (Co-PI) received funding for the project. AK, RWW, PV, CvdH, NEW, EJMB, JK, SSMR, JJvB, TP, ES, FB, HR, EK, PC, SS and RMR designed the study. AK drafted the main part of the manuscript, NW and LdK assisted in writing parts of the manuscript under supervision of AK. EJMB, JL, LdK, SSMR, JJvB, CvdH, ES, FB, HR, EK, PC, SS, RMR, PV and RWW provided critical feedback for important intellectual content on earlier versions of the manuscript. AK, SSMR, NEW, JK, EJMB and JJvB coordinated the data recruitment and data collection. All authors approved the final version of the manuscript for submission and all authors agreed to be accountable for all aspects of the work.

**Funding** This work was supported by a grant from the University of Amsterdam (Spui 21, 1012GC,+31205251400) awarded to AK (Co-PI), PV (PI; p.vonk@uva.nl;

Oude Turfmarkt 151, 1012LA, Amsterdam, University of Amsterdam), CvdH (Co-PI) and RWW (PI; r.w.h.j.wiers@uva.nl; Nieuwe achtergracht 129, 1001NK, Amsterdam, University of Amsterdam). Award/grant number non-applicable. A steering group from the University of Amsterdam monitors the processes of the entire trial and approves mid-term evaluation.

**Competing interests** None declared.

**Patient consent for publication** Not applicable.

**Provenance and peer review** Not commissioned; externally peer reviewed.

**ORCID iDs**
Anke Klein http://orcid.org/0000-0002-0914-0996
Eirini Karyotaki http://orcid.org/0000-0002-0071-2599
Pim Cuijpers http://orcid.org/0000-0001-5497-2743

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
