## [Reviewer comments · BMJ Open]

ARTICLE DETAILS

TITLE (PROVISIONAL)	Online Computer- or Therapist-guided Cognitive Behavioral Therapy in University Students with Anxiety and/or Depression: Study Protocol of a Randomized Controlled Trial
AUTHORS	Klein, Anke; Wolters, N. E.; Bol, E. J. M.; Koelen, J.; de Koning, L.; Roetink, S. S. M.; van Blom, J. J.; Pronk, T.; van der Heijde, Claudia; Salemink, Elske; Bolinski, Felix; Riper, Heleen; Karyotaki, Eirini; Cuijpers, Pim; Schneider, S.; Rapee, Ronald M.; Vonk, Peter; Wiers, Reinout

VERSION 1 – REVIEW

REVIEWER	Saravanan, Coumaravelou University of Sharjah, Family medicine and behavioural science
REVIEW RETURNED	31-Mar-2021

GENERAL COMMENTS	This protocol type manuscript focuses on two types of interventions with the control group. E-intervention is becoming famous after COVID-19. This study will be highly beneficial for the students, therapists, and student counselors. This manuscript will be beneficial for readers if they address the following issues. The title of the research is very difficult to understand so make it very simple for the readers to understand. Abstract: • Under the method section third line 'care as usual' is not clear• Under method section "post intervention at 8 weeks after randomization (t3)". You may delete the 8 weeks after randomization as this will confuse the readers. Second, the manuscript does not show when the intervention started after the randomization of participants. Introduction: • Page 7, lines 52 and 53 some errors in the hypothesis.• The readers will be benefitted if the authors write the detailed aim of this study which is consistent with the hypothesis. Method section: • Does this study inform the participants that their participation is voluntary, participation in this study is not of their course/ program requirement, and withdrawal from this study at any time will not affect their status of the study? if your answer is yes, please mention it in the method section.• Why did the author had used two similar outcome scales PHQ and CES-D to screen depression? Refer to the abstract and method section.• Why the authors did not consider using the Beck Depression Inventory (BDI) as it screens most of the primary and secondary
--

	symptoms of depression compared with PHQ-9. The authors could explain the reason for using PHQ and CES-D than other scales.  • How did you ensure that during intervention participants did not receive treatment or additional counseling from others.? • The manuscript does not mention the participants who exhibited anxiety and/ or depression in the preliminary assessment were not clinically interviewed to confirm the diagnosis because self-reported tools are not sufficient to confirm the diagnosis of depression and anxiety. Before treatment is initiated, participants should be clinically interviewed to confirm their diagnosis. Self-reported assessment alone not warrant for diagnosis of depression and /or anxiety. Second I am not sure MINI scale is sufficient to confirm the diagnosis of depression and /or anxiety after the preliminary assessment. • 'Post assessment after 8 weeks of randomization is not clear. Post assessment should be specific from the first assessment or beginning of the first session of the intervention. • Manuscript does not show how many participants require for each group. • Please mention the secondary outcome are measured before initiation of intervention or not. • Social anxiety will affect the treatment and aggravate depression and anxiety. So how will you manage participants who exhibit social anxiety • Under intervention subtopic second paragraph” The intervention is based on basic cognitive behavioral therapy principles for anxiety and depression with components of psychoeducation and motivational interviewing, including online exercises and home assignments”. Theoretical components of CBT are different from Motivational interviews. (MI) Seven sessions and one booster session is not sufficient to provide CBT and MI. So authors have to mention the objective and outcome of each therapeutic session. You may refer to this manuscript Saravanana,C., Alias,A., Mohamad,M. https://doi.org/10.1016/j.jad.2017.05.037 • A brief description of the therapist-guided vs computer-guided information is necessary for the readers to understand the difference between the two therapy concepts. • How the therapy differs between participants with suicidal and non-suicidal ideas is not mentioned in the manuscript.
--	--

REVIEWER	Nakagawa, Atsuo Keio University Hospital, Clinical and Translational Research Center
REVIEW RETURNED	21-Apr-2021

GENERAL COMMENTS	BMJ Open 2021-049554 Online Cognitive Behavioral Therapy with Therapist versus Automated Feedback in University Students with Anxiety and/or Depression: Study Protocol of a Randomized Controlled Trial The present study protocol aims to test the effectiveness of a therapist-guided versus a computer-guided transdiagnostic iCBT
--

program, focusing on anxiety and depression. In general, the protocol is well written. I have minor points for clarification.

Comments:

1) Background:

All the participants will be enrolled at the University of Amsterdam. Please give a brief description of the University of Amsterdam for international readers.

2) Eligibility criteria:

Participants are excluded from the study if they currently receive psychological treatment for anxiety and/or depression. Please define psychological treatment in this protocol.

3) Intervention:

'iCare Prevent' has been developed and has provided digital therapy to participants and patients. Are there any case series or pre-post study that evaluated the feasibility of this program before conducting this RCT?

VERSION 1 – AUTHOR RESPONSE

Reviewer: 1

The title of the research is very difficult to understand so make it very simple for the readers to understand.

We have adjusted the title to make it better understandable for the reader:

“Online Computer- or Therapist-guided Cognitive Behavioral Therapy in University Students with Anxiety and/or Depression: Study Protocol of a Randomized Controlled Trial”

Abstract:

Under the method section third line 'care as usual' is not clear

We have added an additional sentence to explain 'care as usual' better:

“Participants in the care as usual condition are informed and referred to conventional care services and encouraged to seek the help they need.”

Under method section “post intervention at 8 weeks after randomization (t3)”. You may delete the 8 weeks after randomization as this will confuse the readers. Second, the manuscript does not show when the intervention started after the randomization of participants.

We have deleted the section '8 weeks after randomization'. In addition, we added more details about the start of the intervention on page 11 of the manuscript:

"On this platform, e-coaches create a personal account for each participant directly following the baseline assessment and randomization. Once the personal account has been created, the participant receives an email with information how to activate the account. Once the account is activated, the participant can start with the intervention."

Introduction:

Page 7, lines 52 and 53 some errors in the hypothesis.

We have corrected the errors in the hypothesis on page 7 lines 52 and 53.

The readers will be benefitted if the authors write the detailed aim of this study which is consistent with the hypothesis.

We have now included the specific aim of the study on page 7 of the manuscript right before the hypothesis of the study.

Method section:

Does this study inform the participants that their participation is voluntary, participation in this study is not of their course/ program requirement, and withdrawal from this study at any time will not affect their status of the study? if your answer is yes, please mention it in the method section.

Thank you for noticing this. The participation was voluntary and did not have any relation with course/program requirement and could not affect their student-status (unlinked data-bases). We have now added this part in the method section on page 9 of the manuscript:

"This information letter explicitly states that participation is voluntary and that participants can withdraw at any time without consequences."

Why did the author had used two similar outcome scales PHQ and CES-D to screen depression? Refer to the abstract and method section.

We screen for depression with the CES-D as this questionnaire is part of the screening battery that is used at the University of Amsterdam and was developed in collaboration with other (inter)national universities to be able to collect multicenter data. We have added this information on page 9 of the manuscript.

“This screening questionnaire takes approximately 20 minutes to complete and also includes other questionnaires related to the mental health of university students. This screening questionnaire was developed in collaboration with other universities to collect multicenter data.”

We decided to use the PHQ-9 during the intervention phase to evaluate depression pre- and post the intervention, as this questionnaire only consists of 9 items, and has comparable outcomes to the CES-D and BDI (e.g., Titov et al., 2011). Moreover, some studies even suggested that the CES-D is better to use in college students than the BDI (Santor et al., 1995). In addition, we use an even shorter version of the PHQ, the PHQ-4 which we use prior to each session. Based on earlier experience with online treatment studies in this population, we decided to keep the questionnaires as short as possible to prevent dropout.

Why the authors did not consider using the Beck Depression Inventory (BDI) as it screens most of the primary and secondary symptoms of depression compared with PHQ-9. The authors could explain the reason for using PHQ and CES-D than other scales.

Please see our answer above.

How did you ensure that during intervention participants did not receive treatment or additional counseling from others?

We explicitly ask the participants whether they currently receive any psychological treatment or additional counseling for their problems during the baseline assessment (this is an exclusion criterium). At the post-treatment measurement, sixth months follow-up and 12-months follow-up, we administer medical service use which also includes questions with regard to treatment and counseling (see also page 14/15 and Table 1). This is a self-report measure, so it might still be that participants did receive other treatment or counseling, but did not report it. We have included this as a limitation of the study (please see page 18 of the manuscript).

“Moreover, participants may not report certain information, for example whether they receive face-to-face treatment or counseling for their complaints during the current study.”

The manuscript does not mention the participants who exhibited anxiety and/ or depression in the preliminary assessment were not clinically interviewed to confirm the diagnosis because self-reported tools are not sufficient to confirm the diagnosis of depression and anxiety. Before treatment is initiated, participants should be clinically interviewed to confirm their diagnosis. Self-reported assessment alone

not warrant for diagnosis of depression and /or anxiety. Second, I am not sure MINI scale is sufficient to confirm the diagnosis of depression and /or anxiety after the preliminary assessment.

We are sorry for the confusion. In the first broad screening phase, participants complete the screening, this is a self-report. In the second part of the screening phase, all selected participants (based on the screening) are interviewed with the MINI clinical interview to confirm their diagnosis. The MINI clinical interview is a widely used psychiatric structured diagnostic interview instrument with good psychometric properties (e.g., van Vliet et al., 2007) including reliability and validity (e.g., Sheehan et al., 1997) and is especially often used in clinical research settings due to its relatively short length. It should be mentioned that having a diagnosis is not an inclusion criterium. We merely conducted this interview 1) to check our exclusion criteria, and to 2) have an indication of the characteristics of our sample. We have reformulated several sentences in the method section on pages 9 and 10 to make the procedure clearer.

'Post assessment after 8 weeks of randomization is not clear. Post assessment should be specific from the first assessment or beginning of the first session of the intervention.

We agree that the wording is not clear. We mean that post assessment is 8 weeks after the baseline assessment. Randomization is on the same day as the baseline assessment, so it is the same time point, but we fully agree that it is not clear. We have changed the wording throughout the manuscript.

Manuscript does not show how many participants require for each group.

In total 276 participants are needed, with a ratio of 1:1:1, meaning that 92 participants are needed per group. We have added this information on page 16 of the manuscript.

Please mention the secondary outcome are measured before initiation of intervention or not.

The secondary outcome measures are indeed measured before the start of the intervention, during the baseline assessment. This information can be found in Table 1.

Social anxiety will affect the treatment and aggravate depression and anxiety. So how will you manage participants who exhibit social anxiety.

There are some studies that found that social anxiety is associated with lower treatment outcome. However, these studies do not have sufficient statistical power to establish this association in a definite way. Furthermore, to the best of our knowledge, this has not been tested with an online

transdiagnostic intervention in a university study population. The aim of the current study is to test whether there is an effect. If this is indeed true, we will add social anxiety as a covariate in the model.

Under intervention subtopic second paragraph” The intervention is based on basic cognitive behavioral therapy principles for anxiety and depression with components of psychoeducation and motivational interviewing, including online exercises and home assignments”. Theoretical components of CBT are different from Motivational interviews. (MI) Seven sessions and one booster session is not sufficient to provide CBT and MI. So authors have to mention the objective and outcome of each therapeutic session. You may refer to

this manuscript Saravanana,C., Alias,A., Mohamad,M. <https://eur04.safelinks.protection.outlook.com/?url=https%3A%2F%2Fdoi.org%2F10.1016%2Fj.jad.2017.05.037&data=04%7C01%7CA.M.klein%40uva.nl%7Cb7dedbc510834f382a3308d91ea17161%7Ca0f1cacd618c4403b94576fb3d6874e5%7C1%7C1%7C637574501023435780%7CUnknown%7CTWFpbGZsb3d8eyJWljojMC4wLjAwMDAiLCJQIjoiV2luMzliLCJBTiI6Ik1haWwiLCJXVCi6Mn0%3D%7C2000&sdata=YTBPgQ8sIW9AbjDmGuj8qMRv28UTGOQB%2FFKQXZ38Ues%3D&reserved=0>

Thank you very much for recommending this paper. As the intervention is fully described by Karyotaki et al., 2019, we have decided to only describe the intervention in general terms. We agree that the term motivational interviewing is confusing and not the best term to use. It would be better to refer to this as goal setting, which is a classic component of most CBT protocols. We have deleted ‘motivational interviewing’ on page 11 in the manuscript. We choose to use this intervention including fewer sessions as we want the intervention to be easily accessible and doable with regard to time investment. The participants in our sample do not actively seek help, but are invited based on a screening questionnaire. Also, participants do not have to be diagnosed with a depression and/or anxiety disorder, the invitations are based on a cut-off score on self-reports. Moreover, a recent review and meta-analysis of Etzelmueller and colleagues (2020) to evaluate the effects of internet-based CBT for anxiety and depression indicated that the mean number of sessions of the iCBT protocols that were included in their study was eight, which is highly comparable to the current protocol. They found effect sizes ranging from Hedges’ $g=0.42-1.88$, with a pooled effect of 1.78 for depression and 0.94 for anxiety studies. Previous meta-analytic research also found no significant association between the number of sessions in individual therapies and the effects of the therapy (Cuijpers et al., 2013).

A brief description of the therapist-guided vs computer-guided information is necessary for the readers to understand the difference between the two therapy concepts.

We describe the difference on page 11 and page 12, but we agree that this information is not fully clear. We have therefore added a subparagraph and have now explicitly state what the differences between the two conditions are. Please find this re-written paragraph on page 12 of the manuscript.

How the therapy differs between participants with suicidal and non-suicidal ideas is not mentioned in the manuscript.

Suicidal ideation was an exclusion criterium for the study. Please find this information on page 8 of the manuscript. In addition, we monitor suicidal ideation throughout the study. If a participant indicates suicidal ideas, we call the participant and refer to regular care. Please find this information on page 13 of the manuscript.

Reviewer 2

Background:

All the participants will be enrolled at the University of Amsterdam. Please give a brief description of the University of Amsterdam for international readers.

We have now given a brief description of the University of Amsterdam on pages 8/9 of the manuscript:

“All participants were recruited at the University of Amsterdam (UvA), The Netherlands. The UvA is a public research university located in Amsterdam. It is one of the largest research universities in Europe with 31186 students enrolled in 2018 and includes seven faculties: Humanities, Social and Behavioural Sciences, Economics and Business, Science, Law, Medicine, and Dentistry. Recruitment for the study is done in two phases: a first broad screening phase for all university students studying at the UvA and a second more specific screening of eligible students who showed interest to participate. In the first broad screening phase, all students enrolled at the UvA receive an invitation via email that are sent centrally from the study platform.”

Eligibility criteria:

Participants are excluded from the study if they currently receive psychological treatment for anxiety and/or depression. Please define psychological treatment in this protocol.

We follow the description as made by the APA. (<https://dictionary.apa.org/psychological-treatment>). We have added this in the manuscript.

Intervention:

'ICare Prevent' has been developed and has provided digital therapy to participants and patients. Are there any case series or pre-post study that evaluated the feasibility of this program before conducting this RCT?

This ICare prevent protocol was developed by Weisel and colleagues and adapted by Bolinski and colleagues and Karyotaki and colleagues for college students. The intervention was based on other effective protocols (Buntrock et al., 2016; Ebert et al. 2018, Ebert et al., 2017; Thiart et al., 2015) and can be seen as a variant of an iCBT protocol for which several studies have proven its effectiveness. We have this information on page 10 of the manuscript.

VERSION 2 – REVIEW

REVIEWER	Nakagawa, Atsuo Keio University Hospital, Clinical and Translational Research Center
REVIEW RETURNED	20-Jul-2021
GENERAL COMMENTS	Online Computer- or Therapist-guided Cognitive Behavioral Therapy in University Students with Anxiety and/or Depression: Study Protocol of a Randomized Controlled Trial The present study protocol aims to test the effectiveness of a therapist-guided versus a computer-guided transdiagnostic iCBT program, focusing on university students with anxiety and depression symptoms. I am happy that the authors have clarified the points which I have raised in the revised manuscript.